# Potential Role and Mechanism of Mulberry Extract in Immune Modulation: Focus on Chemical Compositions, Mechanistic Insights, and Extraction Techniques

**DOI:** 10.3390/ijms25105333

**Published:** 2024-05-14

**Authors:** Zaheer Abbas, Yucui Tong, Junyong Wang, Jing Zhang, Xubiao Wei, Dayong Si, Rijun Zhang

**Affiliations:** State Key Laboratory of Animal Nutrition, College of Animal Science and Technology, China Agricultural University, Beijing 100193, China; zaheerabbas@cau.edu.cn (Z.A.); 15956910334@163.com (Y.T.); wangjy9722@163.com (J.W.); zhangjing123@cau.edu.cn (J.Z.); weixubiao01@126.com (X.W.); dayong@cau.edu.cn (D.S.)

**Keywords:** mulberry, bioactive components, anti-inflammatory, antioxidant, immune modulation

## Abstract

Mulberry is a rapidly growing plant that thrives in diverse climatic, topographical, and soil types, spanning temperature and temperate countries. Mulberry plants are valued as functional foods for their abundant chemical composition, serving as a significant reservoir of bioactive compounds like proteins, polysaccharides, phenolics, and flavonoids. Moreover, these compounds displayed potent antioxidant activity by scavenging free radicals, inhibiting reactive oxygen species generation, and restoring elevated nitric oxide production induced by LPS stimulation through the downregulation of inducible NO synthase expression. Active components like oxyresveratrol found in *Morus* demonstrated anti-inflammatory effects by inhibiting leukocyte migration through the MEK/ERK signaling pathway. Gallic and chlorogenic acids in mulberry leaves (ML) powder-modulated TNF, IL-6, and IRS1 proteins, improving various inflammatory conditions by immune system modulation. As we delve deeper into understanding its anti-inflammatory potential and how it works therapeutically, it is crucial to refine the extraction process to enhance the effectiveness of its bioactive elements. Recent advancements in extraction techniques, such as solid–liquid extraction, pressurized liquid extraction, superficial fluid extraction, microwave-assisted extraction, and ultrasonic-assisted extraction, are being explored. Among the extraction methods tested, including Soxhlet extraction, maceration, and ultrasound-assisted extraction (UAE), UAE demonstrated superior efficiency in extracting bioactive compounds from mulberry leaves. Overall, this comprehensive review sheds light on the potential of mulberry as a natural immunomodulatory agent and provides insights into its mechanisms of action for future research and therapeutic applications.

## 1. Introduction

The mulberry plant has a long history of cultivation, particularly in Asia, Africa, Europe, and other continents, due to its high biological activities and medicinal qualities, including antioxidative, immune-stimulating, improved vision, liver protection, and blood pressure lowering effects [1,2]. Furthermore, mulberry is a multipurpose plant and has been recognized as a functional food due to its high nutritive value and phytochemical contents [3,4]. The most well-known *Morus* species are white mulberry (*Morus alba*), black mulberry (*Morus nigra*), and red mulberry (*Morus rubra*) [5,6]. Among them, *M. alba* is the predominant native species, with its root, bark, fruits, and leaves holding significant nutritional and medicinal values. Mulberry species are found in tropical, subtropical, and temperate climates across the world. However, the plant is more common in Asian nations like China, Japan, Korea, and India [7,8]. Mulberries, as fresh fruits, are edible and collected for food production, including juice, jam, and jelly. Mulberry leaves are essential in the sericulture sector since they are the sole source of nutrition for silkworms [8,9]. Because of the beneficial influence on milk production, the leaves are also used for dairy animal feed [10]. Herbal tea prepared from mulberry leaves is popular in Asian nations as a nutritious beverage [4]. Mulberry is an herb known for a long history of use in traditional medicines to cure many diseases and their complications. The bioactive compounds in mulberry leaves are commonly extracted using various methods, including water and ethanol. Various studies have revealed that mulberry leaves are rich in alkaloids, flavonoids, polysaccharides, and phenolic acids, which contribute to their pharmacological properties [11,12].

Mulberry has been shown to possess anti-inflammatory properties [13]. A study published in the journal *Molecules* discovered that the ethanol extract of white mulberry fruits, as well as its derived fractions, contained significant amounts of total phenolic and flavonoid compounds. The findings indicated that the ethanol extract potentially mitigates reactive oxygen species (ROS), diminishes oxidative stress, and suppresses phosphorylation in the mitogen-activated protein kinase (MAPK) pathway by downregulating p38, c-Jun N-terminal kinase (JNK), and extracellular signal-regulated kinase (ERK) expression, while rutin may solely achieve this through p38 inhibition in the MAPK pathway. Moreover, the study indicated a significant reduction in Malondialdehyde (MDA), increased superoxide dismutase (SOD) enzyme levels, and Glutathione (GSH) levels, which is a potent cellular antioxidant [10,14]. Furthermore, treatment with the extract and its fractions restored elevated nitric oxide (NO) production induced by LPS stimulation, primarily by downregulating the expression of inducible NO synthase [13,15]. Another study revealed that *Morus alba* (*M. alba*), as well as its active compound oxyresveratrol, exerted anti-inflammatory effects by inhibiting the migration of leukocytes through the MEK/ERK signaling pathway. Oxyresveratrol derived from *M. alba* inhibited the MEK/ERK kinase activation mediated by C-X-C chemokine receptor (CXCR4) in T cells without affecting CXCR4 receptor expression, thus indicating its targeting of intracellular proteins downstream of chemokine receptor antagonists and potentially serving as a natural anti-inflammatory remedy by inactivating the mitogen-activated extracellular signal-regulated kinase/extracellular signal-regulated kinase (MEK/ERK) pathway, which is integral to modulating inflammatory responses and is a promising target for anti-inflammatory drugs. The study further demonstrated that oxyresveratrol inhibited stromal cell-derived factor (SDF)-1-mediated phosphorylation of the ERK1/2 kinases but enhanced the SDF-1-mediated phosphorylation of p38, suggesting the suppression of CXCR4-mediated chemotaxis via the inactivation of ERK signaling pathway [16,17]. Various extracts of mulberry have been studied and were found to be significant immune modulators and efficient against anti-inflammatory diseases in vivo and in vitro. Mulberry includes alkaloids that protect the body from different threats by activating the immune system following the activation of macrophages. The 1-deoxynojirimycin (DNJ), 1,4-dideoxy-1,4-amino-D-ribitol, and 1,4-dideoxy-1,4-imino-D-arabinitol are the most significant alkaloids among them [18,19]. A recent in vitro study demonstrated that extracts obtained through high hydrostatic pressure were effective in suppressing the release of nitric oxide synthase 2 (*NOS2*) messenger ribonucleic acid (mRNA). Moreover, these extracts were found to reduce the levels of inflammatory cytokines like IL-6 and TNF-α in RAW264 cells induced with LPS. Mulberry leaves ethanoic compound substantially lowers proinflammatory mediators and cytokine production by modifying LPS-induced activation of macrophage cells by reducing nuclear factor-jB (NF-jB) activation [16]. In the LPS-stimulated RAW264.7 macrophage cell line, ethanoic extract of *M. alba* stem at concentrations of 20 and 40 mg/mL inhibits NO production by reduction of both the protein and *iNOS* mRNA [20]. According to scientific research, mulberry leaves have a variety of pharmacological properties and are thought to contain several bioactive substances [3,21]. However, there is currently little evidence indicating that mulberry leaves reduce cardiometabolic risks. The majority of earlier investigations also concentrated on *M. alba*. There has been little research that has focused on the other mulberry species. Therefore, this review aims to assemble and describe the chemical compositions, biological characteristics, and mechanistic insights of mulberry leaves that are connected to the amelioration of health-related risks.

In light of the current literature related to mulberry bioactive components, it will be interesting to review the potential of mulberry in preventing inflammatory-related diseases and discuss their therapeutic mechanism as an immunomodulatory and therapeutic agent against inflammatory diseases. Therefore, this review aims to discuss the therapeutic potential of mulberry in the prevention and control of inflammatory diseases through immune system modulation.

## 2. Biological Portrayal of Mulberry

Mulberry trees, typically deciduous or medium-sized woody perennials, can grow to heights of 10–13 m. They feature upright fissured bark and cylindrical stems that release a milky sap [4,22]. The leaves come in various shapes and sizes, usually measuring 5–7.5 cm in length and 6–10 cm in width. These leaves are deeply lobed, with serrated edges, short acuminate tips, and acute apexes. Their base can be cordate or truncate, and they possess three basal nerves with lateral nerves that split near the edges [23]. The flowers are yellowish-green colored, with a chromosome number of 2n = 28. Female spikes are oval and have stalks, while male spikes (catkins) are broad and cylindrical, with male catkins generally longer than female ones. Botanically, mulberry fruit is a collection of small fruits arranged lengthwise around the central axis, resembling blackberries or loganberries. Known as a syncarp, the fruit consists of numerous drupes enclosed in a fleshy perianth. These drupes are oval or nearly spherical, reaching lengths of up to 5 cm, and can vary in color from white to pinkish-white, turning purple or black when ripe [19]. Morphologically, the mulberry ovary is unicellular with a bifid stigma, similar to other fleshy drupaceous fruits [24].

## 3. Chemical Composition of Mulberry and Its Nutraceutical Perspective

Mulberry is high in macro- and micronutrients, as well as organic acids. Over the decades various phytochemical studies have identified major chemical components of the mulberry associated with significant pharmacological effects [25]. The moisture and ash of mulberry leaves range between 8.19–12.63% and 72.16–79.35%, respectively, according to chemical composition analysis [22,26]. In general, the leaves are high in protein. Mulberry leaves contain significantly more protein than other green leafy vegetables, as well as ascorbic acid and minerals, with calcium and potassium being the two abundant elements, while sodium is present in lesser amounts [21,27]. Previous studies also found anti-nutritional components in mulberry leaves, including fiber, cyanide, and tannins, at concentrations of 8.74–13.70%, 1.01–2.14 mg/kg, and 3.54–5.32 mg/kg, respectively. In addition, mulberries contain a variety of organic acids, including citric acid, tartaric acid, malic acid, succinic acid, and fumaric acid, with malic acid being the main organic acid in all species [28]. Additionally, mulberries are rich in essential minerals such as calcium, phosphorus, potassium, magnesium, and sodium. However, mineral composition may vary among different mulberry phenotypes [29]. Despite belonging to the same genus, *M. alba*, *M. nigra*, and *M. rubra* exhibit differences in physicochemical characteristics. Furthermore, the bioactive compounds in mulberries can also vary depending on several factors, including the species of mulberry, growing conditions, soil composition, climate, and cultivation factors [30]. While there may be some similarities in the types of bioactive compounds found in mulberries across different countries, there can also be significant variations. Differences in soil composition and climate can affect the nutrient content and composition of the mulberry grown in different countries. For example, mulberries grown in regions with rich, fertile soil may also have higher concentrations of certain nutrients compared to those grown in less favorable conditions. Similarly, variations in climate, such as temperature and sunlight exposure, can influence the production of bioactive compounds in the fruit [31,32]. Cultivation practices such as the use of pesticides, fertilizers, and other agricultural inputs can also impact the composition of bioactive compounds in the fruit. Organic cultivation methods may result in higher levels of certain compounds compared to conventionally grown mulberries [32,33]. There may be some similarities in the bioactive compounds found in mulberries across various countries. Overall, variations in growing conditions, cultivation practices, and genetic factors can lead to differences in the compositions and concentration of these compounds. Table 1 below provides a comparative analysis of the different physiochemical parameters of mulberry fruits of different varieties.

### 3.1. Alkaloids

Amino sugars and polyhydroxy alkaloids are active compounds found in mulberry that offer significant health benefits [39,40]. Mulberry’s amino sugars include 1-deoxynojirimycin (DNJ), 1,4-didoxy-1,4-amino-D-arabinitol (DAB)1,4-dideoxy-1,4-amino-2-0-beta-D-glucopyranosyl-deoxynojirimycin, 2,0-alpha-D-glactopyranosyl-deoxynojirimycin (GAL-DNJ) D-fagomine (FAG), norttopaline, and N-methyl-DNJ (N-Me-DNJ); DNA extracted from *M. alba* leaves has shown anti-infective properties by effectively inhibiting the growth of streptococcus mutants through anti-adherence activity [40,41]. In a recent mouse study, it was observed that DNJ extracted from M. alba leaves displayed notable neuroprotective effects at a dose of 160 mg/kg/day. The DNJ effectively inhibited the expression of β-secretase and decreased the β-amyloid buildup, reduced levels of neuroinflammatory markers (IL-1β, IL-6, and TNF-α), and activated the brain-derived neurotropic factor/tyrosine kinase receptor signaling pathway in the brain [42]. Furthermore, DNJ derived from mulberries impeded the proliferation of B16F10 melanoma cells by modulating the activities and expression of matrix metalloproteinase (MMP)-2/9, signifying the anti-metastatic properties against cancerous cells [43]. In a separate investigation conducted in a mouse model with colorectal cancer induced by azoxymethane dextran sodium sulfate, the DNJ from *M. alba* leaves demonstrated a dose-dependent reduction in tumor occurrence and quantity by regulating proapoptotic *BAX* mRNA expression and suppressing anti-apoptotic *Bcl-2* mRNA expression [44]. Furthermore, variations in contents could be attributed to factors such as different varieties, genetics, environments, ecologies, and harvest conditions. The 1-DNJ levels peak in June or July, and the nitrogen application positively influences its contents. Extraction and separation of 1-DNJ from mulberry leaves using 732 resins under appropriate conditions yield promising results. The recovery rate and the purity recorded for the 1-DNJ using 732 resins were 90.15% and 15.3%, respectively [40].

### 3.2. Flavonoids

Flavonoids constitute a crucial functional component of mulberry leaves, comprising approximately 1–3% of the dry weight [45]. Mulberry leaves’ flavonoid improved mitochondrial function and regulated glucose by activating AMPK-PGC1 signaling pathways, thus mitigating hepatic and renal damage and showing antioxidant activity [20,46]. Furthermore, a recent study discovered that mulberry leaf flavonoids improved the dysmetabolism of high fat-induced lipids in mice via intestinal microbiota [11,47]. Mulberries, specifically the *M. nigra* species, are rich in anthocyanins, a subclass of flavonoids that contribute to the vibrant colors of flowers and fruits, ranging from red to blue and purple. Cyanidin-3-glucoside and cyanidin-3-rutinoside are the two major anthocyanins found in mulberries [11,19]. Research has shown that black mulberries have higher levels of total monomeric anthocyanins compared to white mulberries (*M. alba. L*.) and Russian mulberries (*M. alba*) [11,48]. Other anthocyanins identified in mulberry fruit include cyanidin 3-O-(6-O-α-rhamnopyranosyl-β-D-glucopyranoside), cyanidin 3-O-(6″-Oa-rhamnopyranosyl-β-D-galactopyranoside), cyanidin 7-O-β-D-glucopyranoside, cyanidin 3-O-β-D-galactopyranoside, and cyanidin 3-glucoside. These anthocyanins have antioxidant properties, making mulberries a promising source for antioxidant studies [49,50,51].

### 3.3. Protein

Mulberries contain a variety of essential and non-essential amino acids such as alanine, arginine, aspartic acid, glutamic acid, glycine, proline, and serine, as well as essential amino acids like isoleucine, leucine, tyrosine, lysine, threonine, tryptophan, valine, methionine, phenylalanine, and histidine and cysteine [52]. Studies have demonstrated that the total amino acids ratio in mulberries surpasses that of high-quality protein sources like fish and milk [27]. Recent studies have identified new amino acid structures in *M. alba*, including morusimic acids A, B, C, and D, through spectroscopic analysis [53]. Mulberry protein has been found to possess immunomodulatory properties by influencing various immune cells like macrophages, T cells, B cells, and dendritic cells [54]. A specific glycol protein in mulberry fruit known as JS-MP-1 has been shown to significantly enhance the proliferation of B and or T cells when administered at a dosage of 125–2000 µg/mL. Furthermore, mulberry protein (MP) has been found to prevent apoptosis by altering the ratio of BCL-2/BAK protein expression. MP has also been found to help in the migration, maturation, and survival of dendritic cells by activating signaling pathways such as MAP kinase and NF-κB, P38, and extracellular protein kinase (EPK). Recent studies have highlighted the immunomodulatory effects of a specific mulberry fruit protein known as JS-MP1 in conjunction with alkaloids on macrophages. Notably, JS-MP-1 has been linked to an increase in the release of inflammatory cytokines such as macrophage inflammatory protein (MIP-1α), as well as cytokines (1L-6 and TNF-α) from macrophage cells [55]. There is a growing interest in exploring novel bioactive compounds that impact the central nervous system. Bioactive peptides derived from nutrition are less likely to provoke an immune response and offer benefits in terms of natural absorption, digestion, and elimination. Their natural composition makes them a potential alternative with significantly fewer adverse effects compared to synthetic anti-inflammatory and antioxidant sources [56]. The diverse range of protein sources from plants, animals, or fungi presents numerous opportunities for identifying and producing new peptides with pharmaceutical effects. Their precise targeting and minimal harmful effects could provide a more secure choice for medical treatments.

### 3.4. Polysaccharides

Previously, a study claimed that mulberry fruits contain carbohydrates such as fructose and glucose, as well as a significant amount of total phenolic content. Their research also indicated that mulberry fruits could be incorporated into a dietary food plan due to the presence of a special polysaccharide called inulin [53,57,58]. The sugar composition of *M. nigra* and *M. rubra* was analyzed, revealing that glucose accounted for 52% of the composition, followed by fructose at 48% [34]. Furthermore, Gundogdu et al. (2018) reported that mulberry fruits have a glucose content approximately eight times higher than sucrose and fructose content about five times higher than sucrose [59]. Additionally, studies have shown that mulberry polysaccharides extracted from *M. alba* fruit exhibit anti-inflammatory and anti-apoptotic properties. It has been found that these polysaccharides effectively reduce pro-inflammatory cytokines such as interleukin (IL)-1b and IL-6 while increasing the levels of the anti-inflammatory cytokine IL-10. This results in the protection of macrophages stimulated by lipopolysaccharide (LPS) from apoptotic cell death by regulating the levels of anti- and pro-apoptotic protein [60].

It is fascinating to know about the potential benefits of mulberry polysaccharides in cancer immunotherapy. The ability of mulberry polysaccharides to induce phenotypic maturation of dendritic cells by upregulating various immune-related molecules and cytokines is crucial for enhancing the immune response against cancer cells [61]. The increased expression of *CD40*, *CD80/86*, *MHC-I/II* molecules, and cytokines like IL-12, IL-1β, TNF-α, and interferon-β suggests that mulberry polysaccharides can effectively activate dendritic cells and promote their maturation. By decreasing the antigen capture capacity of dendritic cells and enhancing allogeneic T-cell stimulation, mulberry polysaccharides can potentially improve the immune response against cancer cells [62,63]. This dual effect on dendritic cells not only primes them for efficient antigen presentation but also boosts their ability to activate T cells, which are essential for mounting an effective immune response against cancer. Overall, the findings from these studies highlight the immunomodulatory potential of mulberry polysaccharides and their promising role as adjuvants in dendritic cell-based cancer immunotherapy [39]. Further research in this area could lead to the development of novel immunotherapeutic strategies utilizing natural compounds like mulberry polysaccharides to enhance cancer treatment outcomes.

### 3.5. Flavanols

Flavanols are significant components present in mulberries and have been discovered to exhibit diverse biological activities against various diseases [3]. Some of the identified flavanols in mulberry fruits include morin, kaempferol hexosylhexoxide, kaempferol hexoxide, kaempferol rhamnosylhexoxide, quercetin hexoxide, quercetin glucuronide and quercetin hexosylhexoxide [64,65]. Furthermore, mulberry fruits contain myricetin, as well as flavanones such as epigallocatechin, gallate, and naringin. Flavanols like catechin, procyanidin B1, and procyanidin have also been identified in mulberry fruits [4,66].

### 3.6. Phenolic Acids

Phenolic compounds found in mulberry fruits include hydroxycinnamic acids such as chlorogenic acid, caffeic acid, p-coumaric acid, o-coumaric acid, and ferulic acid [67]. Other derivatives of benzoic acids, such as gallic acid, vanillic acid, syringic acid, protocatechuic acid, and p-hydroxybenzoic acid, have also been identified [68]. These phenolic acids have been studied for their therapeutic properties. Mulberries are also known to contain alkaloids, nitrogen-containing compounds with pharmacological activity. Five pyrrole alkaloids were observed in *M. alba*, and similar alkaloids were found in mulberry fruits. In addition, researchers have discovered five new nor tropane alkaloids. Some of these alkaloids have applications in the preparation of anticholinergic drugs [69]. Phenols have garnered increasing attention from researchers and food manufacturers due to the growing recognition of their physiological functions. These abundant micronutrients found in the human diet are believed to play a potential role in preventing various diseases linked to oxidative stress [70]. Researchers have isolated phenolic compounds from mulberry plants such as mulberries, resveratrol, oxyresveratrol, maclurin, and moracin, representing the key active components of the *Morus* species. Various studies have shown that these phenolic components offer numerous health benefits, including anti-inflammatory, antioxidant, and anti-proliferative properties [8,71]. The composition and concentration of phenolic compounds within mulberry leaves vary depending on factors such as variety, cultivation methods, and ripeness and processing techniques. Research indicated that mulberry harvested in May exhibit the highest phenolic content. Furthermore, the enzyme-assisted extraction technique emerges as an efficient and eco-friendly approach for the isolation of phenolic compounds from mulberry leaves, achieving a high extraction yield of 30.0 g/100 g [72]. Detailed information on the physiochemical characteristics of mulberry leaves from different varieties can be found in Table 2.

## 4. Drawbacks of Bioactive Compounds beyond Health and Potential Solutions

While bioactive compounds such as phenolics and flavonoids in mulberry and other sources have numerous health benefits, they may also present certain disadvantages or challenges. Here are some common issues associated with these compounds and potential solutions. One challenge is the limited bioavailability of these compounds, meaning that only a small fraction of ingested compounds are absorbed into the bloodstream and exert their beneficial effects. To enhance the bioavailability, various strategies can be employed, such as encapsulation in lipid-based delivery systems, nano-formulations, or complexation with other molecules like cyclodextrins or proteins. These approaches can improve solubility, stability, and absorption, thereby enhancing the bioavailability of phenolics and flavonoids [78,79]. Phenolic and flavonoid compounds can be prone to degradation during processing, storage, and digestion, leading to loss of bioactivity. Encapsulation techniques, such as microencapsulation or nanoencapsulation, can protect these compounds from degradation by providing a barrier against external factors such as oxygen, light, heat, and enzymes. Additionally, optimizing processing conditions and choosing appropriate storage methods can help minimize degradation and preserve the bioactivity of these compounds [80,81]. Some phenolic and flavonoid compounds possess a bitter taste, which may affect consumer acceptability, especially in food products. Masking techniques, such as encapsulation, microencapsulation, or incorporation into formulations with other ingredients that can mask or balance the bitterness, can be employed to improve palatability without compromising the bioactivity of these compounds [82]. Although phenolics and flavonoids are generally considered safe at moderate levels, excessive intake may lead to adverse effects such as gastrointestinal discomfort or allergic reactions in susceptible individuals. Therefore, it is essential to determine safe dosage levels and consider factors such as individual tolerance, health status, and concurrent medication use. Furthermore, selecting sources with lower levels of potentially toxic compounds or employing purification techniques can help mitigate the risk of toxicity [83,84].

By addressing these challenges through innovative formulation and processing techniques, it is possible to harness the health benefits of phenolic and flavonoid compounds while overcoming their disadvantages, ultimately promoting their utilization in various functional foods, supplements, and pharmaceuticals.

## 5. Extraction of Mulberry Bioactive Components

The increasing demand for functional or medicinal supplements containing natural plant bioactive compounds has led to a quest for effective methods to obtain plant extracts enriched with these components [85]. Previous studies on mulberry leaf extraction have highlighted the advantages of water and ethanol extraction methods due to their operational simplicity and cost-effectiveness, commonly used for extracting alkaloids, flavonoids, polysaccharides, and phenolic acids [63]. The process of water extraction commonly includes the use of distilled water or ethanol to remove impurities, followed by meticulous drying with constant stirring in ultrapure or distilled water at a suitable temperature. Following extraction, the samples undergo filtration, centrifugation, collection, and freeze-drying for further usage [86]. Ethanol extraction involves drying and grinding clean mulberry leaves. It also includes continuous stirring in an aqueous ethanol solution at room temperature, filtering the extracts under reduced pressure using a rotatory evaporator to remove the solvents, and eventually freeze-drying [87]. It is important to note that ethanol is primarily utilized for extracting flavonoids, while water extraction is the preferred method for obtaining polysaccharides. However, variables like temperature, extraction duration, ratio of solid to liquid, and selection of solvent unavoidably influence the composition and biological characteristics of the extract. Among the extraction methods tested, including Soxhlet extraction, maceration, and ultrasound-assisted extraction (UAE), we found that UAE demonstrated superior efficiency in extracting bioactive compounds from mulberry leaves. Various studies showed that using a solvent such as ethanol or methanol in combination with UAE yielded higher yields of bioactive compounds compared to other methods, as shown in Table 3 [88,89]. This finding is consistent with other studies indicating that UAE enhances the extraction efficiency of phenolic compounds from plant materials. Furthermore, the choice of solvent also significantly influenced the extraction efficiency. Solvents with higher polarity, such as ethanol, were more effective in extracting polar compounds like flavonoids and phenolic acids from mulberry leaves [90,91].

Based on the literature, we recommend using ultrasound-assisted extraction with ethanol as the solvent for extracting bioactive components from mulberry leaves. However, it is essential to consider factors such as the target compounds and desired properties of the extract when selecting the extraction method and solvent [92]. Figure 1 illustrates the different extraction and separation techniques.

### 5.1. Enzyme-Assisted Extraction (EAE)

EAE is indeed a promising method for improving the extraction of phytochemicals from plant matrices. By using enzymes to break down cell wall structures, EAE can enhance the release of bioactive compounds that are otherwise trapped or bound within the plant cells. This process not only increases the extraction yield but also helps retain the original properties of the compounds being extracted. The use of enzymes such as cellulase, β-glucosidase, xylanase, β-gluconase, and pectinase can effectively degrade the polysaccharides in the plant cell walls, making the phytochemicals more accessible for extraction [93]. EAE is considered eco-friendly and non-toxic, as it reduces the reliance on organic solvents in the extraction process. Overall, EAE is a valuable approach for optimizing the extraction of phytochemicals from plant materials, especially those that are tightly bound within the cell structures. It offers a more sustainable and efficient method for obtaining bioactive compounds while preserving their original characteristics [94].

### 5.2. Superficial Liquid Extraction (SFE)

SFE is an advanced and environmentally friendly technique that utilizes supercritical fluids at their critical points between the vapor and liquid phase to extract specific elements from solid or liquid materials like plants and food leftovers. This approach commonly uses safe organic solvents to reduce pollution, ensure precise extraction, and quick processing, and maintain the potency of active components without deterioration. By sidestepping prolonged exposure to extreme heat and air, SFE generates residues-free products [95]. The success of SFE hinges on the unique properties of the fluids used, including density, diffusivity, dielectric constant, and viscosity. Adjusting parameters like pressure and temperature allows for the creation of a supercritical fluid (SF). Currently, carbon dioxide stands out as the preferred fluid for SFE due to its ability to exist in a state between gas and liquid, where the density resembles that of a liquid while the viscosity resembles that of a gas. This supercritical state enables SF to exhibit superior transport properties compared to liquid solvents, as the density of SF can be customized by manipulating pressure and temperature. SFE has the potential to become a commonly used extraction method for analyzing botanical, food, and agricultural samples, such as mulberry, due to the growing public interest in the purity and safety of natural products [96]. Radojkovic and colleagues (2016) successfully utilized supercritical carbon dioxide (CO_2_) to extract bioactive compounds from the leaves of *M. alba* and *M. nigra*. Their study revealed that the efficiency of the SFE technique was 1.15 times higher than the traditional Soxhlet method when isolating non-polar compounds using non-polar solvents like hexane [97]. In a separate experiment, SFE was employed to extract α-amyrin acetate from the *M. alba* root bark. The extraction yield increased with temperature when the pressure exceeded 20 MPa, reaching its peak at 40 °C and pressures below 15 MPa. Ultimately, the highest yield of α-amyrin acetate, at 3.68 ± 0.32 mg/g, was achieved at 60 °C and 20 MPa [98].

### 5.3. Pressurized Liquid Extraction (PLE)

PLE enables rapid extraction of compounds while minimizing the use of solvents. This method involves applying high pressure to keep the solvents in liquid form at elevated temperatures, often surpassing the boiling point of the solvents. The elevated temperatures generally enhance the solvents’ ability to dissolve solutes, improve solute diffusion rates, break down solute-matrix bonds, and reduce solvent viscosity and surface tension. However, the application of PLE in extracting phytochemical constituents raises specific concerns [99]. Compared to traditional solvent extraction methods, pressurized liquid extraction (PLE) offers significant advantages in terms of reduced extraction time and solvent usage when extracting bioactive compounds from mulberry. Various factors within PLE, such as the type and ratio of solvent, extraction pressure, and temperature, as well as the number and duration of extraction cycles, can influence the efficiency of extraction [100]. In a separate study, different solvents were evaluated for their effectiveness in extracting total phenolic compounds and antioxidant properties from *M. atropurpurea* fruit using PLE. The results indicated that blends of acidified methanol and acetone in water-organic solvents exhibited enhanced antioxidant activities compared to water or pure organic solvents. Another investigation explored the impact of temperature, extraction duration, and number of cycles on the PLE extraction of rutin and quercetin from mulberry [101]. A refined PLE technique demonstrated superior extraction efficiency in a shorter time frame with minimal solvent usage when compared to ultrasonic-assisted extraction (UAE) and heat reflux methods. For example, PLE achieved more effective extraction of total phenolic compounds from *M. nigra* (2186.09 µg/g vs. 1916.37 µg/g) while using less methanol (47.2% vs. 76.0% *v*/*v*), shorter extraction times (10 min vs. 60 min), and lower temperatures (48 °C vs. 75.5 °C) compared to UAE [100].

### 5.4. Solid Phase Extraction (SPE)

Traditional extraction methods, as well as newer techniques such as SLE, MAE, and PLE, lack selectivity and often lead to the co-extraction of unwanted components like lipids, sterols, and chlorophylls [102]. This complexity can pose challenges for subsequent processes like separation, purification, and analysis using techniques such as GC or HPLC. Therefore, supplementary purification processes are frequently necessary before performing gas or liquid chromatography examinations on the extracts. Combining extraction with sample preparation in solid-phase extraction (SPE) simplifies on-site sampling and analysis, making it a favored method for extracting bioactive compounds from mulberry [103]. In a study, aroma compounds were extracted from *M. nigra*, *M. macrouna*, and *M. alba* fruits using SPE. These findings showed that benzaldehyde, ethyl butanoate, 1-hexanol, hexanal, methional, 3-mercaptohexyle acetate, and 3-mercapto hexanol exhibited significantly elevated odor activity values in comparison to (2-methylthio) ethanol, methional, dimethyl sulfide, and 3-methyl thiophene compounds [104]. In addition to SPE alone, combining SPE with other extraction techniques has proven to be effective for purifying and isolating active components from mulberry plants. For example, Pothinuch and Tongchitpakdee (2011) demonstrated the efficacy of integrating ultrasonic liquid extraction (ULE) with SPE for fractionating melatonin from mulberry leaves of an unknown variety. This combined approach yielded a higher recovery rate (76%) compared to a process that involved homogenization combined with liquid–liquid extraction (12% recovery) [105].

### 5.5. Solid-Liquid Extraction (SLE)

SLE is a widely used method for isolating bioactive components from solid matrices, utilizing organic solvents or water based on polarity differences. The choice of solvent plays a crucial role in the SLE process. Recently, researchers have focused on extracting bioactive components from mulberry using SLE, showcasing its potential [106]. Various studies explored the impact of extraction solvent on *M. alba* fruit polyphenols’ profile and their antioxidant properties against human hepatoma HepG2 cells [107]. A separate study investigated the effect of various factors on the yield of *M. alba* leaf polysaccharides, finding that optimal conditions led to enhanced solubility and yield [63]. Despite its cost-effectiveness and ease of operation, SLE has drawbacks, such as the use of toxic solvents and the need for solvent removal, posing health and environmental risks. To overcome these limitations, new methods are being developed and improvements in traditional methods are taking place to enhance the efficiency and safety of bioactive compound extraction.

**Table 3 ijms-25-05333-t003:** This table shows the technique used for the extraction of various bioactive compounds.

Bioactive Component	Extraction Method	Solvent	Temperature	Duration	Reference
Resveratrol	Soxhlet extraction	Ethanol	60 °C	6 h	[108]
Anthocyanins	Ultrasonic extraction	Methanol	40 °C	30 min	[109]
Flavonoids	Maceration	Water	Room temp.	24 h	[110]
Phenolic compounds	Supercritical CO_2_ extraction	CO_2_	50 °C	2 h	[111]
Polysaccharides	Hot water extraction	Water	90 °C	2 h	[63]
Vitamins	Enzyme-assisted extraction	Cellulase	50 °C	4 h	[112]

## 6. Dietary Significance of Mulberry

Fully ripe mulberry fruits are renowned for their delectable taste, enticing aroma, and flavorful profile. With their succulent and sweet flesh, they can be savored as a delightful snack or incorporated into diverse culinary creations [11]. The nutritional value of mulberry fruits stems from their abundant reserves of vitamins, minerals, and antioxidants. Notably, they boast high levels of vitamin C, iron, and dietary fiber, rendering them a wholesome and nourishing option for dietary consumption. In addition, mulberry fruits are rich in various essential nutrients that play a crucial role in human metabolism [15]. The fruits of *M. alba* are a valuable source of carbohydrates, lipids, proteins, vitamins, minerals, and fibers. The protein content in fresh *M. alba* fruits is higher than that of raspberries and strawberries and comparable to blackberries. Moreover, they contain a higher number of anthocyanins compared to blackberries, blueberries, blackcurrants, and redcurrants [113]. *M. alba* fruits also contain both essential and non-essential amino acids. The essential amino acid to total amino acid ratio is 42%, which is similar to protein-rich foods like fish and milk. Therefore, they can be considered an excellent source of protein [5]. Each variety of *Morus* species, especially *M. nigra*, holds a substantial volume of vitamin C. The content of ascorbic acid in *M. alba* and *M. nigra* is 15.81 and 12.81 mg/100 g of fresh fruit weight, respectively [48].

## 7. Role of Mulberry in Food Industry

The understanding of the relationship between diet and health has led to a significant change in people’s eating habits and lifestyles. This shift in consumer awareness has driven the production of food products that not only meet basic dietary requirements but also provide health benefits [8,114]. Mulberry fruits are well-known worldwide for their delicious taste, making them suitable for consumption, either fresh or as an ingredient in various value-added products and for culinary purposes [4,115]. The growing consumer interest in healthy and low-calorie foods has contributed to the popularity of mulberries and increased demand in the food processing industry [114]. When the mulberry fruits are fully ripe, they are harvested by gently shaking the trees. However, due to their highly perishable nature, mulberries are often underutilized. Nevertheless, there is potential for value addition through various methods [116]. Mulberries are rich in polyphenols that support well-being and can be enjoyed in various ways, including fresh or processed forms like juices, syrups, liquors, molasses, jams, wines, and soft drinks. These superfood berries are highly regarded and offer the potential to create a range of commercially valuable edible products. Numerous patents have been submitted for the medicinal use of *Morus* species, including its potential for lowering blood sugar levels, neurodegenerative disorders, reducing lipid levels, inhibiting tyrosinase, and the development of value-added products [3]. Mulberries have great potential in the fruit and vegetable industry, as they can be used to make a variety of products such as marmalade, jams, jellies, cakes, breads, fruit teas, and fruit drinks. They can also be used in dried, frozen, or fresh forms to produce syrups, wines, and vinegar. Mulberry seeds can be used to extract oil [4,66]. In Turkish cuisine, mulberries are used in traditional foods like “Pestil” and “köme”, which are made from a mixture of mulberries, walnuts, hazelnuts, honey, and flour [117]. In Persian culinary traditions, mulberries are utilized to create jellies, desserts, and sausages, with unripe fruits being used for chutney preparations. Mulberry juice can be stored under cold conditions for up to three months, while bottled juice can remain fresh at room temperature for six months to a year [76]. It is believed that mulberry juice contributes to healthy and smooth skin; aids in preventing irritations, inflammations, and throat infections; and possesses laxative properties. The juice is also used medicinally for various conditions such as fever, cold, malaria, diarrhea, and amoebiasis [3,118]. In China, mulberries are often consumed in a past form called “Sangshengao”, which can be dissolved in warm water to make a tea that is said to improve kidney and liver functions, as well as enhance vision and hearing. In certain Chinese regions, young mulberry leaves and shoots are used as vegetables [119]. In Iran, dehydrated mulberries are used as a natural sweetener in black tea [120].

Nano- and microencapsulation are advanced techniques used in the fields of pharmaceuticals, food science, and agriculture, among others. In the context of mulberry-related research, these techniques have been applied to enhance the delivery, stability, and efficacy of bioactive compounds present in mulberries. Nano- and microencapsulation are used to encapsulate bioactive compounds within small particles, such as nanoparticles or microspheres, to protect them and enhance their functionality. Nano-encapsulation involves encapsulating bioactive compounds, typically with a size range of 1–100 nanometers [121]. This technique offers several advantages, including increased stability, improved solubility, controlled release, and target delivery of the bioactive compounds. To carry out nano-capsulation, various methods can be employed, such as nanoprecipitation, emulsion solvent evaporation, and electrospinning [78]. Nanoencapsulation has been employed to encapsulate bioactive compounds found in mulberries, such as anthocyanins, flavonoids, and polyphenols. These compounds exhibit various health benefits, including antioxidant, anti-inflammatory, and anti-cancer properties. By nano-encapsulating these compounds, researchers aim to enhance their stability during storage and improve their absorption and bioavailability in the body. Another technique called microencapsulation involves encapsulating bioactive compounds within larger particles with a size range of 1–1000 μm. This technique is useful in protecting bioactive compounds from degradation, controlling release kinetics, making taste or odor, and improving handling and processing properties. Common methods for microencapsulation include spray drying, fluidized bed coating, and coacervation [122]. Microencapsulation has been utilized to encapsulate mulberry extracts or individual bioactive compounds for various applications, including functional foods, nutraceuticals, and pharmaceuticals. By microencapsulating these compounds, researchers can mask undesirable tastes or odors, protect them from environmental factors, and control their release kinetics in the body. Overall, nano- and microencapsulation hold great promise for enhancing the efficacy and applicability of mulberry bioactive compounds in various health-related applications, including functional foods, dietary supplements, and pharmaceutical formulations.

## 8. Medicinal Properties of Mulberry

Mulberry has diverse therapeutic attributes stemming from its chemical composition, which imbues it with anti-inflammatory, antioxidant, antimicrobial, hepatoprotective, and other beneficial properties. Table 4 outlines an overview of research findings on the medicinal uses of the mulberry plant. The section below presents a review of the existing literature on how the mulberry plant’s composition contributes to enhancing human health and overall wellness through its medicinal properties.

### 8.1. Anti-Inflammatory Effect of Mulberry

To put it simply, when microorganisms enter certain tissues and circulate in the blood, they can cause inflammation, which is a complex vascular response in the body. If left unchecked, inflammation has the potential to escalate into chronic conditions such as rheumatoid arthritis, cardiovascular diseases, and cancers [140]. Nevertheless, research indicates that the consumption of natural plants processing anti-inflammatory properties can be beneficial in addressing persistent inflammation. Recently, a new extraction technique called high hydrostatic pressure was used to extract compounds from *M. alba*. These extracts were found to inhibit the release of nitric oxide and the expression of genes associated with inflammation in laboratory cells [141]. Extracts from the root bark of *M. alba* were found to block the production of nitric oxide in cells stimulated by lipopolysaccharide [142]. Additionally, extracts from *M. alba* leaves were found to reduce the production of proinflammatory mediators and cytokines by suppressing the activation of macrophage cells. Similarly, extracts from the stem of *M. alba* showed anti-inflammatory activity by inhibiting the production of nitric oxide in laboratory cells stimulated by lipopolysaccharide [143,144].

The ethanolic extract of *M. alba* leaves has been found to effectively reduce the production of proinflammatory mediators and cytokines by regulating the activation of macrophage cells induced by lipopolysaccharide (LPS). The results further suggested that the ethanol extract could reduce reactive oxygen species (ROS), lower oxidative stress, and inhibit phosphorylation in the MAPK pathway by reducing the expression of p38, JNK, and ERK. Rutin, on the other hand, might achieve this by specifically inhibiting p38 in the MAPK pathway. Additionally, the study observed a notable decrease in MDA levels and an increase in SOD enzyme and GSH levels, indicating enhanced cellular antioxidant activity [16]. This is achieved through the suppression of nuclear factor-jB (NF-jB) activation. Additionally, the ethanolic extract of *M. alba* stem has exhibited anti-inflammatory activities in the murine RAW364.7 cell line by inhibiting the production of nitric oxide (NO) and suppressing the iNOS, both protein and mRNA levels [143]. Furthermore, studies have shown that white mulberry root extract has strong antihistamine and anti-allergic properties, inhibiting compound 48/80-induced systemic allergic reactions and histamine release both in vitro and in vivo. Root extract also inhibits mast cell-mediated allergic reactions [145]. These findings suggest that mulberry species have potential as a natural source for the development of anti-inflammatory drugs. In addition, the comprehensive effects of mulberry leaves in anti-inflammatory and antioxidant activities are summarized in Table 3.

A study was conducted to evaluate the protective effects of DNJ, extracted from the leaves of the *M. alba* plant, on cognitive decline, β-amyloid accumulation, and neuroinflammation in senescence-accelerated prone mice. The results indicate that administering DNJ at a dose of 160 mg/kg per day effectively suppresses the activity of β-secretase, decreases the deposition of β-amyloid, reduces neuroinflammation markers (IL-1β, IL-6, and TNF-α), and enhances the brain-derived neurotrophic factor/tyrosine kinase receptors pathway in the brain. Consequently, DNJ demonstrates potential as a protective agent for the nervous system, offering the possibility of alleviating pathological alterations in the brains of individuals with Alzheimer’s disease [42]. Additionally, studies have shown that DNJ extracted from mulberry trees has the potential to impede the spread of B16F10 melanoma cells by potentially reducing MMP-2/9 activity and presence, increasing *MMP-2* mRNA expression, and alternating cell surface binding properties. These results highlight the anti-diffusion properties of DNJ concerning melanoma cells [43]. In another study, the effect of DNJ from mulberry trees on colorectal cancer induced by azomethane dextran sulfate sodium was investigated. These findings lead to the conclusion that DNJ effectively reduces tumor incidence and volume in a dose-dependent manner by enhancing the mRNA expression of pro-apoptotic *BCL-2* [146]. Furthermore, DNJ alleviated the growth inhibition of epithelial cells by increasing the mRNA expression of lymphoid enhancer factor-1, a key gene involved in regulating mammary gland growth and development [147].

### 8.2. Immune Modulatory Effect of Mulberry

Mulberry leaf extract exhibits various biological properties, including immunomodulatory and antioxidant effects. The impact of mulberry leaf polysaccharides (MLP) on the composition and characteristics of gut microbiota for immune modulation has been studied recently. The investigation showed that MLP significantly affects the diversity, immune-related factors, and levels of short-chain fatty acids (SCFAs) in the gut microbiota, as shown in Figure 2. The findings revealed that MLP was able to improve thymus and spleen indexes, repair the damaged intestinal barrier, reduce inflammatory cytokine levels in cyclophosphamide (CTX)-treated mice, and alleviate oxidative damage in the liver. Furthermore, MLP was associated with an increased abundance of Bacteroidetes but decreased levels of Firmicutes, Butyricimonas, and Eubacterium in CTX-treated mice. Additionally, MLP restored the diminished levels of acetic acid, propionic acid, and n-butyric acid caused by CTX. These investigations suggest that MLP, as a unique component, may modulate the immune response by influencing the composition of gut microbiota and SCFA production [148,149]. Derived from *M. alba* fruit and leaf, mulberry polysaccharides have been found to influence the development of dendritic cells, which are crucial in initiating immune responses. Furthermore, the polysaccharides extracted from the fruit of *M. alba* can act as an immune modulator. They can activate murine RAW264.7 macrophage cells, prompting the release of chemokines such as RANTES and macrophage inflammatory protein-1 α, along with pro-inflammatory cytokines such as TNF-α and IL-6 [150]. Furthermore, they can induce the production of inducible nitric oxide synthase (*iNOS*) and cyclooxygenase (COX)-2, which play a role in the production of nitric oxide [151]. The study concluded that polysaccharides from mulberry sources could be used as an immune therapeutic and a helpful health supplement for regulating the immune system.

Mulberry fruit protein (MFP) possesses immunomodulatory properties that have been investigated on various immune cells such as T cells, B cells, dendritic cells, and macrophages. One specific glycoprotein found in mulberry fruit, known as MP, has been shown to significantly enhance the proliferation of T and/or B cells when administered at a dosage of 125–2000 μg/mL [66]. Additionally, mulberry protein has been found to prevent apoptosis by altering the ratio of Bcl-2/Bak protein expression. MFPs also play a role in the maturation, migration, and survival of dendritic cells by activating signaling pathways such as MAP kinase and NF-κB, as well as p38 and extracellular protein kinase (EPK) [152]. Further research has indicated that a specific mulberry fruit protein, JS-MP-1, along with pyrrole alkaloids, exhibits immunomodulatory effects on macrophages. In particular, JS-MP-1 has been associated with heightened release of inflammatory chemokines (MIP-1α and RANTES) and cytokines (IL-6 and TNF-α) by macrophage cells (murine RAW264.7), whereas pyrrole alkaloids boost the generation of IL-12, TNF-α, and nitric oxide [55].

### 8.3. Antioxidant Activities of Mulberry

Mulberries have been found to possess high nutritional value and have attracted the attention of scientists who have studied their antioxidant properties. Scientists have examined the phenolic makeup and antioxidative properties of *M. alba* fruit, discovering a notable relationship between antioxidative activity and the overall phenolic content [11,26]. The antioxidant activities have been highly favored with the ripening of the fruit. Additionally, studies have shown that fully ripened mulberry fruits have higher antioxidant efficacy, especially when air-dried [153]. Black mulberry has also been found to have high levels of antioxidant activity. Mulberry extract using methanol has shown strong antioxidant activity. Similarly, extracts of black mulberry using ethanol, ethanol–water, and water have also demonstrated high antioxidant efficacy [154].

Anthocyanins found in mulberry fruits have been studied for their antioxidant and anti-fatigue properties. Purified anthocyanins from mulberry juice and mulberry mark have been found to act as antioxidants and help reduce fatigue [155]. Comparative studies have shown that mulberry fruits have higher resveratrol content and antioxidant activity compared to other fruits such as Jamun and Jackfruit. Resveratrol, present in mulberries, has been found to reduce reactive oxygen species levels and suppress lipid peroxidation, thus reducing the chances of DNA damage [16,156]. Overall, these studies highlight the exceptional antioxidant properties of mulberries, making them a valuable addition to one’s diet.

There is a body of evidence demonstrating the antioxidant activity of mulberry fruit protein (MFP) fractions (MFP1, MFP2, MFP3, and MFP4). MFPs exhibit antioxidant properties by utilizing the carboxyl/carbonyl group to facilitate the binding of peroxy radicals to hydrogen atoms, thereby halting radical chain reactions. The selenyl or seleno-acid ester group activates the anomeric carbon of MFPs, leading to an enhanced antioxidant capacity. Among the four MFPs, MFP-4 contains a significant portion of low-molecular-weight fractions and higher levels of galacturonic acid, contributing to its superior DPPH scavenging activity [57].

### 8.4. Antimicrobial Effect of Mulberry

Recent studies have been investigating the antimicrobial properties of natural plants and their components. One plant, *M. alba,* contains a substance called chalcomoracin that has been shown to have antimicrobial effects against methicillin-resistant Staphylococcus aureus [157]. Researchers have also discovered that a hydro-methanolic extract from the stem bark of *M. alba* exhibits antimicrobial properties against various bacteria, including *Enterococcus faecalis*, *Staphylococcus epidermis*, *Escherichia coli* (*E. coli*), and *Salmonella typhimurium*. Furthermore, an ethanolic extract from mulberry trees has been found to effectively inhibit the growth of *Propionibacterium acnes* and *Staphylococcus epidermis* [158]. In a separate study, bioactive components were isolated from *M. alba* fruit using column chromatography, and its structure was identified using NMR and spectroscopy. The extracted compound showed antibacterial activity against *streptococcus mutans* [39].

These findings highlight the potential of *M. alba* and *M. nigra* as sources of antimicrobial compounds, which could be utilized in the development of novel therapeutic agents against bacterial infections [6,159]. These findings suggest that various parts of the *Morus* tree, including its juice, vinegar, and other value-added products, have demonstrated potential antimicrobial properties [160]. For example, *M. nigra* juice showed antimicrobial potential against bacteria such as *B. spizizenii* and *P. aeruginosa*. Vinegar produced from *M. alba* exhibited antibacterial effects against a range of bacteria, including *S. aureus*, *S. pyogenes*, *E. coli*, and *B. cereus*, as well as antifungal activity against *C. albicans* [161]. Among these bacteria, *S. aureus* and *S. pyogenes* showed the highest zone of inhibition, while *E. coli* exhibited the lowest zone of inhibition. These findings further highlight the potential of the *Morus* tree as a source of antimicrobial agents [162].

### 8.5. Anti-Inflammatory Effect of Mulberry through Regulation of Gut Microbiota

Several research findings highlight the pivotal role of gut microbiota in maintaining the integrity of the intestinal barrier, a crucial element in shaping and regulating immune function. The gut microbiota can exert either immunosuppressive or immunoprotective effects by generating substances that are either anti-inflammatory or pro-inflammatory, respectively [148]. Intestinal inflammation can be triggered by the movement of bacteria from the gut, as shown where *E. coli* cultures induce the release of proinflammatory cytokines in mucosal samples from patients with inflammatory bowel disease (IBD) [163]. However, certain types of bacteria are beneficial, such as *Lactobacillus*, *faecalbacterium*, and *Bifidobacterium*, which have been discovered to reduce inflammation in the gut by decreasing the expression of pro-inflammatory cytokines and increasing the anti-inflammatory substances. This beneficial effect may be due to the bacterial by-products that impact the microbial function. The human gut contains a diverse range of bacteria and microorganisms, and an imbalance in the gut microbiota can lead to a decrease in beneficial microbes and an increase in harmful pathogens. This imbalance can contribute to metabolic disorders, oxidative stress, and inflammation, affecting overall health [164,165].

Active compounds found in mulberry have been shown to help restore the balance of bacterial communities associated with conditions like oxidative stress, inflammation, and metabolic disorders. These compounds are also shown to inhibit pathways related to metabolic disorders like insulin resistance and improper cell function. Specific microbial communities, such as *Bacteroidetes* and *Firmicutes,* play a crucial role in energy metabolism and cellular physiological functions [148,166]. Ongoing research has revealed that certain mulberry compounds like flavonoids, polysaccharides, alkaloids, and phenolic acids can enhance gut microbiota diversity in animal models with certain metabolic disorders, as shown in Table 3. These compounds can influence the composition of gut microbiota, increasing the beneficial bacteria like *Lactobacillus* and *Bacteroidetes* while reducing harmful bacteria [167]. Among the key products of microbial fermentation in the gut are short-chain fatty acids (SCFAs), notably acetate, propionate, and butyrate, which contribute significantly to the normal functioning and structure of the gut and colonic epithelial cells. Polysaccharides have been shown to influence the composition of gut microbiota, fostering the presence of beneficial species, such as SCFAs-producing bacteria, while suppressing potential pathogens, thus contributing to overall health maintenance [168]. In a specific study, the administration of cyclophosphamide (CTX) led to a significant reduction in the levels of acetic, propionic, and n-butyric acids in the fecal [148] contents of mice, alongside an increase in bacteria associated with immunosuppression, such as Butyricimonas and Eubacterium. Conversely, treatment with mulberry polysaccharides resulted in elevated SCFA levels and a decrease in these bacteria. These findings suggest that mulberry leaf polysaccharides have the potential to modulate CTX-induced alterations in gut microbiota, thereby potentially influencing the immune response.

### 8.6. Mulberry against Cancer

Many naturally occurring substances have been found to exhibit anticancer effects by triggering apoptosis (cell death) in tumor cells and halting the cell cycle, which is considered an effective strategy to combat irregular cell growth [169]. Mulberry plants have been long known for their medicinal properties, with the potential to inhibit cell proliferation, likely attributed to the presence of flavonoids, known for their effectiveness against certain types of cancers [167]. Several studies have highlighted the role of mulberry in preventing cancer in animal models. A purified lectin from *M. alba* leaves, known for its anti-proliferative properties, has been shown to induce apoptosis in human breast cancer (MCF-7) and colon cancer (HCT-15) cells by triggering significant morphological changes and DNA fragmentation associated with apoptosis [170]. The root bark of *M. alba* contains flavanone glycoside, 5,2′-di-hydroxyflavanone-7,40-di-O-b-D-glucoside (steppogenin-7,4′-di-O-b-D-glucoside), which has anti-proliferation activity against HO-8910 cells in human ovarian cancer [171]. Another study investigated the use of a Moracin N (MAN) probe to label and identify protein targets in lung cancer cells. MAN showed a growth-inhibitory effect on these cells and targeted the programmed death ligand 1 (PD-L1) checkpoint pathway and T-cell receptor (TCR) signaling pathway, suggesting an immune-regulatory function. Cell-free surface plasmon resonance (SPR) results confirmed MAN’s direct interaction with PD-L1 protein. Molecular docking analysis revealed MAN’s binding to a specific residue of PD-L1 protein. MAN downregulated PD-L1 expression and disrupted PD-L1/programmed death 1 (PD-1) binding. Co-culture experiments showed MAN’s ability to increase CD8+ Granzyme B (GZMB)+ T cells and decrease CD8+PD-1+ T cells, indicating an anti-cancer effect through blocking the PD-L1/PD-1 signaling. In vivo experiments demonstrated that MAN combined with anti-PD-1 antibody inhibited lung cancer development and metastasis synergistically. Overall, secondary metabolites of mulberry leaves target the PD-L1/PD-1 signaling, enhance T cell-mediated immunity, and inhibit lung cancer tumorigenesis, potentially improving the efficacy of immune checkpoint inhibitors in lung cancer treatment [172].

## 9. Conclusions

Mulberry, particularly *M. alba*, has been found to possess various bioactive compounds that can positively impact immune function. These compounds exhibit anti-inflammatory, antioxidant, and immunomodulatory properties that contribute to overall immune system regulation. This review paper emphasizes the importance of understanding the chemical compositions of mulberries and how they interact with the immune system to exert their beneficial effects. Furthermore, the modulation mechanisms involved in mulberry’s immune-enhancing properties are discussed, shedding light on the potential pathways through which mulberry can support immune function. Overall, this review paper underscores the promising role of mulberry in immune modulation and calls for further research to fully elucidate its mechanisms and potential applications in immune-related conditions.

In the future, research on the potential role and mechanism of mulberry extract in immune modulation is poised to make significant advancements in understanding its effects on the immune system. Further exploration of the chemical compositions of mulberry extract and their interaction with immune cells could provide valuable insights into the specific compounds responsible for its immunomodulatory properties. By elucidating the mechanistic pathways through which mulberry extract exerts its effects on immune function, researchers may uncover novel targets for therapeutic intervention in immune-related disorders. Additionally, advancements in extraction techniques for obtaining bioactive components from mulberry extract could enhance its efficacy and bioavailability for potential clinical applications. Collaborative efforts between scientists, pharmacologists, and clinicians may lead to the development of innovative mulberry-based immune modulators with enhanced potency and specificity, offering promising therapeutic options for immune-mediated diseases in the future.

## Figures and Tables

**Figure 1 ijms-25-05333-f001:**
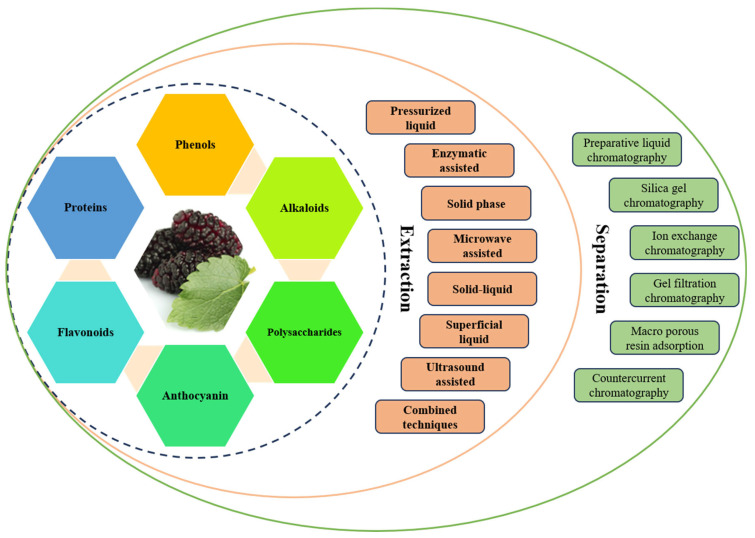
This figure shows various extraction and separation techniques used for bioactive compounds from mulberry.

**Figure 2 ijms-25-05333-f002:**
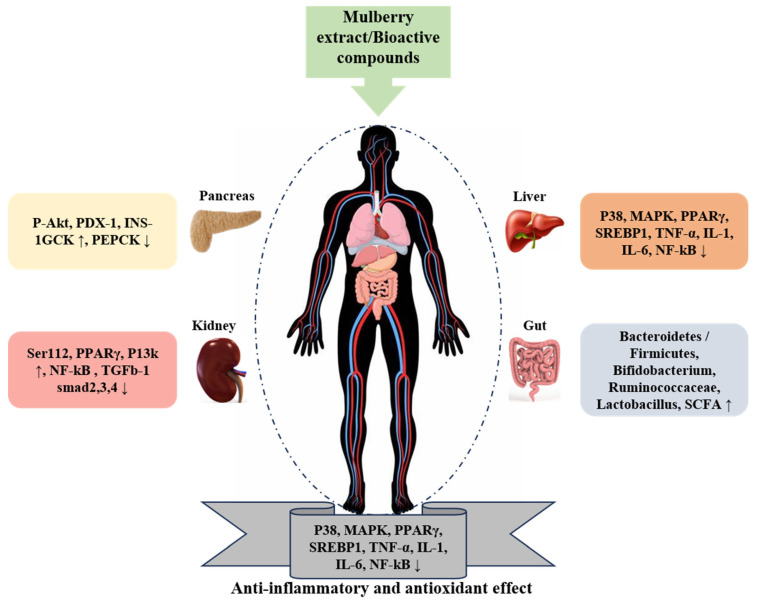
Demonstrates the significant role of mulberry as an anti-inflammatory, antioxidant, and gut microbiota regulator. The upward arrow in the figure shows the increase while the downward arrow shows the decrease.

**Table 1 ijms-25-05333-t001:** This table shows the bioactive components of various mulberry fruit species.

Physiochemical Characteristics	*M. alba*	*M. nigra*	*M. rubra*	Reference
Total dry matter (g)	29.5	27	24.4	[34]
Moisture (%)	71.5	72.6	74.6	[35]
Ash (g/100 g)	0.57	0.50	2.45	[35]
pH	5.6	3.52	4.04	[34]
Protein (g/100 g)	1.55	0.96	1.2	[36]
Glucose (g/100 g)	6.864	7.748	6.068	[36]
Fat (%)	1.10	0.95	0.85	[34]
Fiber (g/100 g)	1.47	11.75	-	[35]
Phenol (mg/100 g)	181	1422	1035	[34]
Flavonoids (mg/100 g)	29	276	219	[34]
Anthocyanin µg/g	911.8	719	109	[37]
Ascorbic acid (mg/100 g)	22.4	21.8	19.4	[34]
Calcium (mg/100 g)	152	132	132	[34]
K (mg/100 g)	1688	922	834	[34]
Fe (mg/100 g)	4.2	4.2	4.5	[34]
Nitrogen (%)	0.75	0.92	0.82	[34]
Malic acid (g/100 g)	3.095	1.323	4.467	[36]
Succinic acid (g/100 g)	0.168	0.342	0.132	[36]
Citric acid (g/100 g)	0.393	1.084	0.762	[36]
Organic acid (g/100 g)	3.983	2.951	5.812	[36]
Fructose (g/100 g)	6.269	5.634	5.407	[36]
Total soluble solids (%)	7.27	11.60	19.20	[38]
Total acidity (%)	0.25	1.40	1.37	[34]

**Table 2 ijms-25-05333-t002:** Physiochemical components of various types of mulberry leaves.

Physiochemical Characteristics	*M. alba*	*M. nigra*	Reference
Moisture %	51.1–59.7	51.3–66.9	[73]
Crude protein (%)	20.94–29.15	27.63–37.36	[74]
Carbohydrates (%)	3.7	3.1	[75]
Crude fiber (g/100 g)	5.1–8.4	3.6–16.61	[76]
Total ash (%)	---	14.78	[77]
Potassium (mg/100 g)	2.1–2.4	1.2–3.9	[73]
Sodium (mg/100 g)	10	10	[73]
Magnesium (mg/100 g)	0.5–0.9	0.4–1.4	
Sulphur (mg/100 g)	0.2–0.3	0.2	
Nitrogen (g/100 g)	2.1–2.3	2.2–3.1	
Calcium (mg/100 g)	1.7–3.2	1.7	
Lead (mg/kg)	0.4–0.8	0.3–0.6	
Carbon (g/100 g)	39–41.4	37.4–41.3	
Titanium (mg/kg)	5.5–10.8	5.4–8.1	
Iron (mg/kg)	119.3–241.8	124.7–197.8	
Zinc (mg/kg)	23.9–39.5	29.5–34.2	
Lithium (mg/kg)	3.1–10.1	1.9–17.2	
Copper (mg/kg)	4.4–5.8	4.2–5.9	
Nickel (mg/kg)	2.1–2.9	1.7–5.4	
Boron (mg/kg)	253.5–825.3	277.4–548.3	
Molybdenum (mg/kg)	0.8–1.3	1.1–2.3	

**Table 4 ijms-25-05333-t004:** This table shows the significant role of mulberry leaves as an anti-inflammatory and antioxidant. The upward arrow shows the increase and the downward arrow shows the decrease.

Mulberry Extracts/Components	Model Used	Extraction Technique	Pharmacological Effect	Mechanism of Action	References
Mulberry leaf extract	Male db/db mice	Hot air mill	Protect Ielts-b cells	Endoplasmic reticulum stress (ERS) ↓, β-cell apoptosis ↓, β-cell proliferation ↓	[123]
Mulberry leaf extract	Male Wistar rats		Antioxidative	Glucose-6-phosphate dehydrogenase (G6PDH), Glutathione peroxidase (GPx), Glutathione–S–transferase (GST), Superoxide dismutase (SOD) ↑, Chloramphenicol acetyltransferase (CAT) ↓	[124]
Mulberry leaf extract	Male Wistar rats	Electric mix	Antioxidative	GPx, Glutathione reductase (GR), GST, SOD ↑, CAT ↓	[125]
Mulberry leaf extract	Male SD rates		Gut microbiota regulation	*Coprobacillus*, *Runinococcaceae*, *Bifidobacterium*, *Bacteroides*, *Prevotella*, *Collinsella* ↑	[126]
Mulberry leaf extract	Male db/db mice	Air flush dried	Antioxidant, anti-inflammatory	Tumor necrosis factor-α (TNF-α), Monocyte chemotactic protein-1 (MCP-1), Cluster of differentiation 68 (CD68), Thiobarbituric acid reactive substances (TBARS), Nicotinamide Adenine dinucleotide phosphate (NADPH) oxidase subunits, PU.1 ↓	[127]
Mulberry leaf extract	INS-1 cells	Ethanol	Induce autophagy	Microtubule-associated protein 1A/1B-light chain 3 (LC3II/1) ↑, Phospho-adenosine monophosphate kinase (p-AMPK), p62, Mammalian target of rapamycin (mTOR) ↓	[68]
Mulberry leaf extract	Male C57BL/6 J mice	Water	Gut microbiota regulation	Short-chain fatty acids (SCFA) ↑, *Bacteroidetes*, *Proteobacteria* ↑, *Firmicutes* ↓	[128]
Mulberry leaf extract	Male Wistar rates	Acetone and ethanol–water	Antioxidant	Liver and kidney Fe ↓, *TBARS* ↓ Liver Cu ↑	[129]
Mulberry leaf extract	C57BL/6 mice	Pressure extraction	Pancreatic β-cells protection	Protein kinase B (p-Akt) ↑	[130]
Mulberry leaf extract	Male ICR rats	Water	Antioxidant and anti-inflammatory	TNF-α; Toll-like receptors (TLR)-2; Myeloid differentiation factor (MyD)-88, TNF receptor-associated factor 6 (TRAF6), Nuclear factor-kappa-B (NF-κB) p65; Insulin receptor substrate (IRS)-1 ↓, Insulin receptor substrate(InsR) ↑	[131]
Mulberry leaf extract	C57BL/6 mice	Ethanol	Antioxidant and decreased liver fibrosis	Lipoprotein lipase (LPL), Sterol regulatory Element-binding transcription factor 1 (SREBP1c), aP2, Liver-X-receptor alpha (LXRa), Fatty acid synthase (FAS), CCAAT/Enhancer-binding protein (C/EBPα) ↓, Uncoupling protein 2 (UCP2) ↑, Alpha-smooth muscle actin (a-SMA), collagen ↓, 4-hydroxynonenal protein adducts(4-HNE), Nuclear factor erythroid 2-related factor 2 (Nrf2), Heme oxygenase-1 (HO-1), GPx ↓	[132]
Mulberry leaf extract	Male SD rats	Ethanol	Antioxidant, anti-inflammatory	Malondialdehyde (MDA), lipid hydroperoxide, DPPH Radical ↓	[133]
Mulberry leaf extract	SD rats	Ethanol	Regulate gut microbiota	*Bacteroidetes*, *Bacteroides*, *Barnesiella*, *Ruminococcus* ↑, *Actinobacteria*, *Bifidobacterium*, *Romboutsia*, *Lactobacillus* ↓	[134]
Mulberry leaf extract	C57BL/6 J mice	Ethanol	Regulate gut microbiota	*Bacteroidetes*, *Firmicutes*, *Bacteroidetes*, *Actinobacteria*, *Epsilonbacteraeota*, *Cyanobacteria*, *Alloprevotella*, *Muribaculaceae*, *Parabacteroides* ↑, *Romboutsia*, *Gastranaerophilales* or *Oscillatoriales_cyanobacterium* ↓	[135]
Coumaric acid	Male Wistar rats	Hydrolysis	Reduce inflammation and fibrosis	Smooth endoplasmic reticulum (Ser)-112, Peroxisome proliferator-activated receptor-gamma (PPARγ), NF-kb ↓	[136]
DNJ and QG	HepG2 cells		Anti-inflammatory	P38, MAPK, PPARγ, SREBP1, TNF-α, interleukin (IL)-1, IL-6, NF-kb ↓	[137]
Polysaccharides	Male Wistar rats	Water	Antioxidants improve insulin sensitivity and suppress renal fibrosis	P13K, p-IRS1, 2, p-Akt-1, p-Akt-2, p85α ↑	[138]
Flavonoids and Alkaloids	Male db/db mice		Repair kidney and live damage	Phosphatidylinositol–3 kinase (P13k) ↑, NFk-b, Transforming growth factor beta (TGFb)-1 SMAD family member (smad)2,3,4 ↓	[139]

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
