# Peer review of "Potential Role and Mechanism of Mulberry Extract in Immune Modulation: Focus on Chemical Compositions, Mechanistic Insights, and Extraction Techniques"

_ijms, 2024, doi:10.3390/ijms25105333_

Round 1
Reviewer 1 Report
Comments and Suggestions for Authors
1.This review stated that it aimed to elaborate the therapeutic potential of mulberry in the prevention and control of inflammatory diseases through immune system modulation. On the other hand most part of the article was dedicated to explain the chemical composition of the mulberry and extraction methods of the bioactive compounds from mulberry. Therefore, either aim of the paper should be revised or the aforementioned part from the article should be summarized and the therapeutic potential of the plant should be explained in more detail (Part 7)
2. It should be useful to include a table that summarize the extraction methods and conditions for the bioactive component from mulberry.
3. Table 3. The mode of action, includes many abbreviation they should be explained within the manuscript to make it more clearer to the reader
4. At the conclusion part, the future studies and the lack of the area in the reported literature on therapeutic effect of mulberry should be addressed
5. The references doesn't cover the recent studies. They dated back to 2022, studies reported in the literature for the last two years should be included..
6. At the abstract it is perceived that the protein content of the fruit is very high, and it creates a wrong perception. On the other hand only the leaf is high in protein, and therefore a correction must be made for that sentence. The abstract part should be revise to include some findings from the literature summurized thrgout out the manuscript
Author Response
Dear reviewer:
We deeply appreciate the time and effort you put reviewing this article. We have studied the comments carefully and have made corrections/responded, which we hope to meet with the approval.

Reviewer 2 Report
Comments and Suggestions for Authors
The article " Potential Role and Mechanism of Mulberry extract in Immune Modulation: Focus on Chemical Compositions, Mechanistic Insights, and extraction techniques" is relevant and interesting. The manuscript could be acceptable after major revisions listed below
1) All the scientific name of the plant and genes must be in italic
2) The abstract should be rewritten and does not clearly express the results of the work, please correct it. Highlights the significant results and finding you present in this review
3) Most of the results from previous experiments that you present are too overall. You must explain their results in detail and indicate clearly for instance
A study pub- 48 lished in the journal Molecules discovered that the ethanol extract of white mulberry 49 fruits, as well as its derived fractions, contained significant amounts of total phenolic and 50 flavonoid compounds!!!!
Another study revealed that Morus alba (M. alba), 56 as well as its active compound oxyresveratrol, exerted anti-inflammatory effects by inhib- 57 iting the migration of leukocytes through the MEK/ERK signaling pathway!!!
Please rewrite all the literature in detail
4) You mentioned mulberry's chemical composition and its nutraceutical potential. Are the bioactive compounds the same in different countries? Please compare and explain why.
5) You mentioned different Extraction protocol for bioactive components of mulberry, So based on the results explain which solvent and which method are the best and you suggested?
6) The bioactive compounds including phenolic and flavonoids and... have some disadvantages, You must indicate and explain how you can solve these problems.
7) It is clear that Mulberry is critical in the food industry. Can you explain the techniques that can be applied to carry bioactive compounds for their better function, such as nano and microencapsulation?
8) The aim of this study is to understand the molecular mechanism of Mulberry extract in immune modulation and other biological potentials, such as anti-inflammatory properties. However, I have not come across a comprehensive explanation of the mechanisms behind these actions. Can you please highlight and explain each mechanism individually?
Comments on the Quality of English Language
The English needs to be checked by the English native person
Author Response

(The authors gave the same response as above.)

Reviewer 3 Report
Comments and Suggestions for Authors
The abstract is nicely formulated but I would add some data, numbers, or tangible fact about the important of Mulberry rather than general information. I would add how this review help other researcher and reader more than supplying information.
Table 1: Units needed for the presented data. For example, the fiber content is 1.47. Is this percentage? g/100 g??
Fig 2: Any permission right
Despite this is purely review but I would see the gaps in RD of product extraction and provide possible future RD.
Comments on the Quality of English Language
Dear Editor,
This is a nice review article. Very important fruit in our life and worth publishing it after considering the reviewers comments.
Author Response

(The authors gave the same response as above.)

Round 2
Reviewer 1 Report
Comments and Suggestions for Authors
Dear authors, thank you very much for reviewing your article comprehensively and correcting it in line with my suggestions. Since this article is a review, its reference list mostly include the recent studies on mulberry reported in literature between the years 2023 and 2024. When you search the search engines, you can see how much work has been done in the last two years. However, in this revision, only 15 of the 164 references belong to the last two years. I think that old references should be replaced with new references or the number of references from the last two years should be increased much more.
Author Response
Dear reviewer:
Thank you for your diligent review of the article and for your thoughtful suggestions. Your feedback underscores the importance of ensuring that the references cited in the review accurately reflect the most recent advancements in the field of mulberry research.
Indeed, the landscape of mulberry-related studies has seen significant evolution in recent years, as evidenced by the wealth of literature emerging between 2023 and 2024. Your observation regarding the distribution of references is valuable, and we appreciate your insight into the need for a more robust representation of recent studies.
In response to your recommendation, we took proactive measures to enhance the currency of the reference list. The new number of current references of 2023 and 2024 have been increased to 37 from 16. This involved replacing outdated references with more current ones or increasing the inclusion of studies from the last two years. By doing so, we aim to ensure that our review provides readers with a comprehensive and up-to-date perspective on the subject matter.
Once again, I extend my gratitude for your constructive feedback. Your input contributes significantly to the refinement and integrity of our work, and we are committed to addressing your concerns in the revised version of the article.
Reviewer 2 Report
Comments and Suggestions for Authors
The authors have been improve the manuscript base on the comments and It can be publish in the present form
Author Response
Dear Reviewer:
Thank you very much for taking the time to review our manuscript and for providing your feedback. We sincerely appreciate your positive assessment of the improvements made to the manuscript based on the previous comments. Your encouragement regarding the readiness of the manuscript for publication in its current form is greatly valued. We are committed to ensuring the quality and clarity of our work, and your input has been instrumental in achieving this goal. Thank you once again for your time and constructive feedback.
Round 3
Reviewer 1 Report
Comments and Suggestions for Authors
Dear authors,
Thank you very much for your response. I am happy to decide that it can be published as it is.